# Fairness without Sensitive Attributes via Noise and Uncertain Predictions

## Abstract

While model fairness improvement has been explored previously, existing methods invariably rely on adjusting explicit sensitive attribute values in order to improve model fairness in downstream tasks. However, we observe the trend of sensitive demographic information being inaccessible as public concerns around data privacy grow. In this paper, we propose a confidence-based hierarchical structure of variational autoencoder (VAE) architectures called "Reckoner" for reliable fairness learning under the assumption of missing sensitive attributes. First, we present the results of exploratory data analyses conducted on the widely-used COMPAS dataset. We observed significant disparities in model fairness across different levels of confidence. Inspired by these findings, we devised a dual-model system in which the model initialised with a high-confidence data subset learns from the model initialised with a low-confidence data subset, enabling it to avoid biased predictions. To maintain predictiveness, we also introduced learnable noise into the dataset, forcing the data to retain only the most essential information for predictions. Our experimental results show that Reckoner consistently outperforms state-of-the-art baselines on both the COMPAS and the New Adult datasets in terms of both accuracy and fairness metrics.

## 1 Introduction

Automated models and algorithms have found wide application in various domains, including finance and justice, as tools to assist human decision-making processes (Con (2022); Brennan et al. (2009)). These applications collect information like age and education level in financial services, or misconduct incidents in policing, as well as sensitive data like race and gender from individuals, raising concerns about the ability of automation to deliver accurate and equitable judgments across diverse demographic groups. Due to concerns about the misuse of private data, increasing regulatory restrictions have made it more challenging to access and make use of sensitive information in automated decision making (Voigt & Von dem Bussche (2017)). Approaches to improve fairness without sensitive data can be roughly divided into two categories: focusing on maximising the utility of the worst-case group (Hashimoto et al. (2018); Lahoti et al. (2020); Wei et al. (2023)) and focusing on limiting the impact of sensitivity-correlated proxy on predictions (Gupta et al. (2018); Zhao et al. (2022); Yan et al. (2020)). By relying on the the correlated observed attributes, these methods have the potential benefit of unfairness mitigation. However, we argue that fairness is still underachieving because the correlated attributes are unreliable.

We believe that approaches to fairness without sensitive attributes face two major challenges. First, relying on proxy combinations is not reliable. Existing methods mostly rely on manually selecting combinations of proxies based on the correlation between them and sensitive attributes or empirical information (Datta et al. (2017); Zhao et al. (2022)). The problem is that only a small subset of proxies rather than all relevant non-sensitive attributes would be considered, which can still allow embedded bias in the data to hinder improvements in fairness predictions. For example, in existing literature, there are almost no methods that treat age as a proxy in tasks with the COMPAS dataset, while in our analysis, we observed an interesting interaction between age and sensitive attributes, such as race. Moreover, manual selection is also highly inefficient, as different task scenarios imply different suitable proxy groups, requiring careful selection of the attributes to be adjusted. Additionally, such methods are difficult to apply to unstructured data and have limited generalisability. The second problem is the trade-off between prediction accuracy and fairness. Like many efforts to

improve fairness (Wick et al. (2019); Feldman et al. (2015)), achieving a win-win situation for both prediction accuracy and fairness remains challenging.

In this paper, we introduce a novel confidence-based framework that ensures both predictability and fairness of classification results, leveraging learnable noise and a knowledge-sharing mechanism between a dual-model system. We first analyse the distribution patterns of selected non-sensitive information across subsets at different classification model confidence levels. We observe that when data is close to the decision boundary, non-sensitive information associated with those data tends to be similarly distributed across demographic groups, leading to lower accuracy but increased fairness. In contrast, when data is far from the decision boundary, non-sensitive information shows varying distributions across demographic groups, resulting in higher accuracy but reduced fairness. Inspired by these findings, we initially divide the original training set into two subsets based on confidence scores obtained from a simple linear classifier and then initialise generators with a Variational Autoencoder (VAE) as the backbone. Then we introduce learnable noise into the original data aiming to retain only the necessary information for prediction. In the next phase, one generator acquires knowledge from the other generator to learn fairness while also updating itself using the ground truth to maintain high levels of effectiveness.

The main contributions of this paper are as follows:

- By analysing the distribution of non-sensitive attributes across demographic groups in different model confidence interval, we observe that biases implicitly contained within non-sensitive attributes hinder the ability of the model to make fair judgments. This analysis also sheds light on the relationship between predictability and fairness at different confidence levels.

- We introduce a novel confidence-based classification framework, named Reckoner. This framework achieves improved fairness while maintaining accurate predictions in classification results by utilising learnable noise and knowledge-sharing in a dual-model system. This provides an effective approach to improving fairness without using sensitive attributes.

- We conduct extensive experiments using real-world datasets to evaluate the effectiveness of the proposed framework as compared to other baselines in terms of fairness and predictive performance. We also present the results of an ablation study to understand the impact on effectiveness of the two main components in this framework.

## 2 RELATED WORK

**Group Fairness.** In contrast to approaches that emphasise the equitable treatment of similar individuals in pursuit of individual fairness, our work focuses on group fairness, manifesting in the differential treatment of distinct demographic groups. Some prevalent methods include incorporating fairness regularisation to the objective function or converting it into a constrained optimisation problem. Kamishima et al. (2011) introduced a method for reducing mutual information between sensitive groups and targets by quantifying the mutual distribution between them. This approach aims to diminish the dependency between sensitive groups and targets. A similar concept is also reflected in Beutel et al. (2019), where fairness is achieved by minimising the absolute correlation between these two entities. In contrast to the aforementioned methods, Hardt et al. (2016) proposes the use of the equalised odds fairness metric, which underscores the equalisation of true positive and false positive rates across different demographic groups. It transforms the general loss function into an optimisation problem subject to fairness constraints, ensuring that the revised unbiased predictions closely approximate the original predictions. Similarly, Zafar et al. (2019) achieves fair classification by adding tractable constraints at the decision boundary. However, as the desire for both algorithmic fairness and privacy grows, we observe the requirement of avoiding the use of sensitive attributes in machine learning model training, leading to legislative restrictions on such practices like, e.g., the General Data Protection Regulation (GDPR)(Voigt & Von dem Bussche (2017)). To manage such requirements, some approaches have been designed under the assumption that sensitive attributes are either difficult to obtain or prohibited from use.

**Fairness Without Sensitive Attributes.** To deal with fairness problem under this setting, the main idea of some studies is leveraging the correlation between sensitive and non-sensitive attributes to mitigate bias. Representative work includes the use of proxy features (Gupta et al. (2018)), in which

a proxy group is obtained from clustering the data and is used to replace actual sensitive groups during training. A well-known example is using 'zip code' instead of 'race' as this can have similar effects on individual splits since the two attributes are highly correlated (Datta et al. (2017)). Similarly, Zhao et al. (2022); Yan et al. (2020) explore features which have strong correlation with sensitive attributes to learn fair classifiers by using them for training and for regularisation in learning. However, this approach needs a careful selection of proxy attributes and even of fairness metrics. To address underlying issues, Zhu et al. (2023) estimates fairness using only weak proxies. Through estimating the transition probabilities between sensitive group target values, it uses auxiliary models to calibrate the fairness metrics. Another family of approaches, Hashimoto et al. (2018), addresses the limits of missing sensitive attributes via techniques taken from distributionally robust optimisation (DRO). The main idea involves utilizing observed correlated attributes to identify regions with low utility and assigning higher weights to individuals within these regions to improve worst-case performance and thus improving fairness. Recently, Jung et al. (2023), targeting group fairness, extends DRO with fairness constraints in the resulting objective function using a re-weighting based learning method. Beside the aforementioned methods, others have recently utilised various techniques to address unfairness without the knowledge of demographics. For example, Lahoti et al. (2020) adversarially reweighs the samples to achieve a Rawlsian Max-Min fairness and learn the task classifier. Others tackle the problem through knowledge distillation (Chai et al. (2022)), reweighing-based contrastive learning (Chai & Wang (2022)) and causal variational autoencoder (Grari et al. (2022)). However, these methods need the prior identification of proxies to harness their interactions with sensitive attributes, such as correlation and causality, in order to achieve fairness. Our approach avoids the need for such analysis. Instead, it leverages learnable noise applied to all data, forcing the data to retain only essential information for better predictions. Additionally, it employs a dual-model knowledge-sharing mechanism to acquire fairness-related knowledge, thereby improving predictive fairness. Hence, our proposed framework exhibits greater generalisability, particularly when dealing with data where proxy identification is challenging, such as images and audio.

## 3 ANALYSIS OF MODEL FAIRNESS BASED ON CONFIDENCE SCORES

**Problem definition.** Our goal is to improve fairness in prediction tasks in a non-sensitive attributes setting, where a set of labeled data $\mathbb{D} = \{x_i, y_i\}_{i=1}^{N}$ is available for training. Each $x_i \in R^{1 \times m}$ is a m-dimensional data instance, and we use $F = \{f_i, \ldots, f_m\}$ to denote the m features. Sensitive attributes S are not used in training, i.e. $S \notin F$. Following the task settings on COMPAS and New Adult datasets (Larson et al. (2016); Ding et al. (2021)), we focus on binary classification problems, i.e., $y_i \in 0, 1$.

We assume that even when sensitive information is excluded, remaining relevant data can still introduce unintended biases and unfair errors. In this analysis, we aim to uncover unintended biases in the COMPAS dataset, which is used for predicting offender recidivism. Figure 1a provides a simple example. In our setup, sensitive information within the red dashed box is omitted from training. Unfairness can arise when individuals of the same demographic and similar circumstances receive harsher judgments or more prior convictions, leading the model to mistakenly view some as more dangerous. Therefore, previous work has focused more on previous misconducts and the severity of the original crime (Zhao et al. (2022); Le Quy et al. (2022)), whereas we are interested in another attribute, age, which is also relevant to the predictions. Figure 1b and 1c show the distribution patterns of the 'Age' attribute and 'Previous Misconduct' in various subsets, partitioned based on confidence scores.

Some confidence-based works, such as OOD (Out-of-Distribution) detection (Hendrycks & Gimpel (2016)) and image classification (Cui et al. (2022); Corbière et al. (2019)), have inspired us to consider that data with varying levels of confidence may hold crucial information for uncovering misinformation. As a result, we divided the training dataset into two subsets based on a confidence threshold of 0.6 as Lakkaraju et al. (2017) suggested, leading to the following findings: (1) Surprisingly, the ages of individuals from different racial backgrounds exhibit distinctly different distribution patterns within these subsets compared to 'Previous Misconduct ' attribute. However, most prior literature has overlooked the age attribute and did not consider it as a proxy for sensitive attributes. (2) The model appears to attempt to capture patterns in the feature distribution for predictions but is misled by the majority. We observe a tendency toward right-skewness in age distribution of both the entire training dataset and

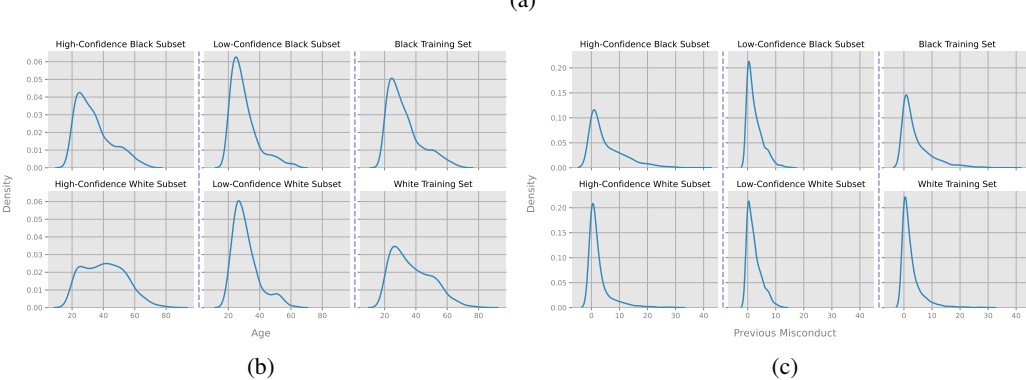

(a)

(b)                                              (c)

Figure 1: (a) An example of the COMPAS dataset. In our experiment, the attribute 'Race' in the red frame is not involved. (b) and (c) Distributions of the attribute 'Age' and 'Previous Misconduct' across different subsets of the training set.

| Subsets | Fairness Metrics(%) | |
| --- | --- | --- |
| | Equalised Odds | Demographic Parity |
| High-confidence Subset | 25.10 | 32.90 |
| Low-confidence Subset | 8.10 | 8.50 |
| Training Set | 20.30 | 24.60 |

Table 1: Results of Equalised Odds and Demographic Parity in different subsets

the high-confidence subset, which comprises approximately 65% of the training data. Additionally, within high-confidence subset, there are varying tendencies of age distribution dispersion and right-skewness among different racial groups. However, These differences are not observed in the low-confidence subset, where the model's performance is notably suboptimal. As mentioned in Barry et al. (2023), this discrepancy may arise from the misleading feature distribution patterns captured by the model. (3) We find that the model tends to provide relatively fair predictions in subsets with lower confidence levels, whereas unfairness in predictions becomes more notable in subsets with higher confidence levels. Table 1 illustrates the variations in the Equalised Odds and Demographic Parity fairness metrics across different subsets. Building on the previous finding, we can infer that the predictions from the model stem from capturing the feature distribution of the majority. However, the exhibited tendency may potentially contain unfair biases and errors, leading to a reduction in fairness.

The above analysis underscores the relationship between fairness and model confidence while further disclosing how non-sensitive information can impact the accuracy and fairness of model predictions.

## 4 METHODS

**Overview.** As shown in Figure 2, our proposed method consists of two training stages, the Identification stage and Refinement stage. In the Identification stage (Sec. 4.2), we employ a simple linear classifier, such as logistic regression, to perform a binary classification task on the raw dataset under a supervised learning setting. Then the training data is split into two subsets based on a predefined confidence threshold: a high-confidence subset and a low-confidence subset. These subsets are then used to initialise their respective generators. At the beginning of the Refinement stage (Sec. 4.3),

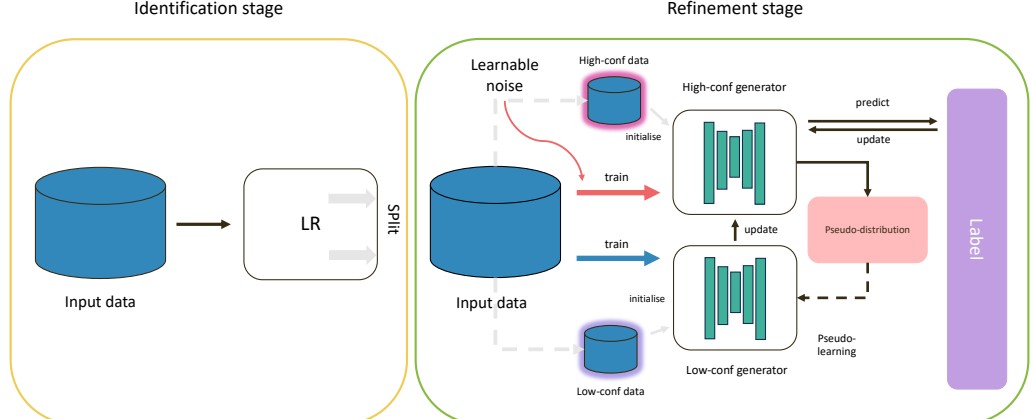

Figure 2: Overview of **Reckoner**. Reckoner consists of two stages. ***Identification stage***: we first train a logistic regression classifier on the raw data, and then split the data based on confidence scores. In ***Refinement stage***, we introduce learnable noise into the original dataset. We employ two generators, one for low-confidence instances and another for high-confidence ones. The *Low-Conf* generator uses pseudo-distribution produced by the *High-Conf* generator for limited training times and restores for each new data. Knowledge acquired during this process is then shared with the *High-Conf* generator, which incorporates ground truth data to refine its model weights.

we introduce learnable noise to the original dataset, generating noise-augmented data for training in this stage. The purpose of using learnable noise is to encourage each augmented input $x_i$ to "neutralise" embedded unfairness and provide only necessary information for prediction. Next, during the iterations, the *High-Confidence* (or "*High-Conf*") generator produces latent vectors to train the *Low-Confidence* (or "*Low-Conf*") generator, and the *Low-Conf* generator updates its knowledge back to the *High-Conf* generator. The *Low-Conf* generator is trained for a limited number of iterations, for example, three epochs, before reverting to its initialised state. This approach ensures good and unbiased prediction while maintaining an efficient training process. Lastly, the *High-Conf* generator uses ground truth and shared knowledge to update its parameters.

### 4.1 MOTIVATION

The motivation behind our proposed framework stems from the analytical findings in Section 3. We believe that manually selecting combinations of sensitivity-correlated attributes and adjusting their relationship with target values is unreliable, as unfairness may be embedded throughout the entire dataset. However, discovering relationships among all attributes is inefficient (Wu et al. (2020)). To enhance fairness and retain predictiveness, it is sufficient to predict only essential information, and we can leverage characteristics from distinct subsets to enable the model to learn how to achieve fair classification. To improve fairness in the settings of missing sensitive attributes, we design a novel framework, named Reckoner, which seamlessly integrates learnable noise and a knowledge-sharing mechanism between dual models. We have demonstrated the necessity of combining these two components in our ablation study (see Section 5.2).

### 4.2 IDENTIFICATION STAGE

In this stage, we perform a simple confidence-based sample split on the training data to obtain high-confidence samples and low-confidence samples. Specifically, we train a logistic regression classifier on the original training set and we split the data using confidence threshold (in this case, it is set to 0.6 following Lakkaraju et al. (2017) ). Our hypothesis is that the model trained on the low-confidence subset is more inclined towards fair classification, even if its predictive accuracy is relatively modest. However, by integrating the knowledge derived from the model trained on the high-confidence subset, it is possible to enhance fairness while maintaining predictive performance.

### 4.3 REFINEMENT STAGE

#### 4.3.1 LEARNABLE NOISE

In the initial phase of this stage, learnable noise is introduced into the training set. From a predictive perspective, the model only needs the necessary information to achieve accurate predictions, while the remaining parts, which may contain unfairness, can be adjusted. This concept aligns with some of the work that utilises disentanglement to enhance fairness (Creager et al. (2019); Park et al. (2021)). Selecting only a subset of relevant information for adjustment is unreliable because some unfairness may be hidden within easily overlooked attributes, such as the age of offenders. Intuitively, learnable noise can assist in mitigating the embedded biases in the training data by forcing the model to focus solely on predicting the most essential information. To be more specific, we add a noise wrapper to vectors of the same dimensions as the input, which are randomly drawn from a normal distribution. The noise wrapper is a simple two-layer MLP that is subsequently applied to modify the input. The resulting modified input is referred to as a noise-augmented input and can be represented as follows:

$$\tilde{x}_i = x_i + g_\omega(\eta), \tag{1}$$

where $\omega$ is the set of parameters in the noise wrapper $g$, and $\tilde{x}_i$ is the new input we use for the *High-Conf* generator in rest of the refinement stage.

#### 4.3.2 DUAL-MODEL AND KNOWLEDGE SHARING

The *Low-Conf* generator, initialised with the low-confidence subset, may not necessarily generate highly informative representations that effectively represent the entire dataset. However, it treats different demographic groups equally, illustrating a special case of the trade-off between fairness and accuracy. The notably poor predictive performance prevents us from relying on the *Low-Conf* generator to perform classification tasks, but it can guide the *High-Conf* generator to make fair classification to some extent. Specifically, within the Reckoner framework, the *Low-Conf* generator relies on the pseudo-distribution generated by the *High-Conf* generator for supervised learning to update its own parameters. Since ground truths are not involved in this phase, the learning process can only be referred to as *pseudo-learning*. This approach helps to improve the *Low-Conf* generator to obtain better parameters for prediction. The supervised loss of *pseudo-learning* consists of three components: the Evidence Lower Bound (ELBO) (Kingma & Welling (2013)), which serves as the objective of the VAE, the regression loss of expectations $\mu_L$ and $\mu_H$, and the regression loss of variances $\sigma_L^2$ and $\sigma_H^2$:

$$\mathcal{L}_L = \mathcal{L}_{VAE}^L + \mathcal{L}_\mu + \mathcal{L}_{\sigma^2}, \tag{2}$$

where $\mathcal{L}_{VAE}(p, q) = \mathbb{E}_{q(z|x)}\left[\log p(x|z)\right] - D_{KL}\left[q(z|x)||p(z)\right]$ is ELBO loss of the vanilla VAE. The subscripts "H" and "L" refer to the *High-Conf* generator and the *low-Conf* generator, respectively. Note that during this phase, the training iterations of the *Low-Conf* generator are limited (set to only 3 times in our experiments) for training efficiency. Furthermore, once these iterations end, the *Low-Conf* generator has a rollback operation, reverting its parameters to their initialised values. This design avoids the acquisition of biases and unfairness inherent in the dataset.

On the other hand, we rely on the *High-Conf* generator, which offers higher precision, to perform classification tasks. However, as revealed by the analysis in Section 3, we are aware of its poor performance in terms of fairness. Previous works (Liu et al. (2021); Herzog & Hertwig (2014)) suggest that averaging the opinions of two independent models provides a more effective enhancement in prediction quality. This concept of leveraging the strengths of both models is prevalent in various research areas, such as the high-pass and low-pass filters in graph neural networks (Bo et al. (2021)) and Exponential Moving Average (EMA). The most promising improvement on fairness for the *High-Conf* generator lies in integrating the knowledge from the *Low-Conf* generator, and its parameter update mechanism can be expressed as follows:

$$\Theta_H \leftarrow \alpha\Theta_H + (1-\alpha)\Theta_L, \tag{3}$$

where $\alpha$ controls the proportion of the knowledge of *High-Conf* generator. In order to enhance predictive accuracy, we use ground truths and employ the backpropagation algorithm to update the *High-Conf* generator. By integrating the knowledge from the *Low-Conf* generator, the final update mechanism can be formulated as follows:

$$\theta_i^H \leftarrow \hat{\theta}_{i-1}^H - \gamma\frac{\partial\mathcal{L}_H}{\partial\hat{\theta}_{i-1}^H}, \quad \hat{\theta}_{i-1}^H \leftarrow \alpha\theta_{i-1}^H + (1-\alpha)\theta_k^L, \tag{4}$$

where $\theta_i^{\mathrm{H}}$ is the weight of *High-Conf* generator at $i$-th iteration, $\hat{\theta}_{i-1}^{\mathrm{H}}$ is the temporary weight integrating both *High-Conf* generator's and *Low-Conf* generator's knowledge controled by $\alpha$, and $k$ is the iteration number when the *Low-Conf* generator achieve the best performance during *pseudo-learning*. The supervised loss of classification task consists of three components: the ELBO loss and the classification loss, which is binary cross entropy:

$$\mathcal{L}_{\mathrm{H}} = \mathcal{L}_{\mathrm{VAE}}^{\mathrm{H}} + \mathcal{L}_{cls}. \tag{5}$$

Intuitively, the *pseudo-learning* applied to the *High-Conf* generator can be interpreted as shifting the predictive distribution closer to the decision boundary, with the hyperparameter $\alpha$ controlling model stability. Another component of the framework, learnable noise, retains only the most predictive aspects of the input, ensuring both accuracy and enhanced prediction fairness. We will discuss the contributions of these two components to prediction fairness in the discussion on ablation study (see Section 5.2).

## 5 EXPERIMENTS

**Datasets.** We validate our model on two benchmark datasets:(1) **New Adult**: The dataset utilised for Adult reconstruction, as introduced by Ding et al. (2021), comprises 49,531 samples, each associated with 14 attributes. In contrast to Adult, New Adult retains the actual income values rather than labels. The primary objective is to predict whether an individual's income exceeds 50k. In our experiments, we convert income into binary labels, and we set race as the sensitive attribute and exclude it for experiments. (2) **COMPAS**: COMPAS (Larson et al. (2016)) comprises 7,215 data samples, each associated with 11 attributes. Follwing previous works on fairness without sensitive attributes (Chai et al. (2022)), we have filtered this dataset to include only African American and Caucasian offenders, hence the modified dataset containing 6,150 samples. The primary objective is to predict whether a offender will commit another offense within two years. We set race as the sensitive attribute and exclude it for experiments.

**Baselines.** We compare our method with four related methods for fair comparisons: (1) Distributed Robust Optimisation (DRO) (Hashimoto et al. (2018)): The primary objective of this method is to enhance Rawlsian Max-Min Fairness (Rawls (2001)), by maximising utility for the worst-case group. It achieves this by using observed attributes to identify problematic regions and assigning higher weights to individuals within those regions. (2) ARL (Lahoti et al. (2020)): This approach also aims at Rawlsian Max-Min Fairness and uses adversary learning to optimise worst-case performance by prioritising instances with higher losses. (3) FairRF (Zhao et al. (2022)): This approach identifies features strongly correlated with sensitive attributes and minimise the correlation by reweighting to achieve fairness. (4) Chai et al. (2022): This approach applies knowledge distillation requiring one model to produce soft labels, and use them to train a second model to obtain a better decision boundary. It has two variants: either with softmax label or with linear label.

**Implementation Details.** For feature engineering, since both datasets we are using in the experiments contain categorical features, we employ feature hashing on categorical features to avoid having sparse training data. For the proposed framework, we apply logistic regression to train a simple binary classifier and follow Lakkaraju et al. (2017) setting 0.6 as the confidence threshold for data splitting in the identification stage. In the refinement stage, we use each confidence-based subset and use 10% of the total model training iterations to initialise both the *High-Conf* generator and the *Low-Conf* generator. In the *pseudo-learning* phrase, the *Low-Conf* generator is trained three times. In the training for the whole proposed framework, we use Adam to be the optimiser, MSE loss for calculating regression loss in the *pseudo-learning* phrase, and binary cross entropy for classification loss. For evaluation, we use Equalised Odds (Berk et al. (2021)) and Demographic Parity (Corbett-Davies et al. (2017)) as fairness metrics and report accuracy for classification. Both fairness metrics are considered better when they have lower values.

### 5.1 RESULTS

Table 2 and Table 3 show the comparison results of our models with other baselines. Note that results of both variants from Chai et al. (2022) and of FairRF(Zhao et al. (2022)) are from Chai et al. (2022) with the same datasets and same train-valid-test split. Our model consistently performs favourably against the baselines. In COMPAS dataset, we can observe that Reckoner achieves the

Table 2: Results on the COMPAS dataset

| Metrics(%) Methods | Accuracy | Equalised Odds | Demographic Parity |
|---|---|---|---|
| DRO (Hashimoto et al. (2018)) | $67.48 \pm 0.21\%$ | $21.01 \pm 1.25\%$ | $25.41 \pm 1.90\%$ |
| ARL (Lahoti et al. (2020)) | $68.72 \pm 0.88\%$ | $20.95 \pm 1.01\%$ | $25.29 \pm 1.31\%$ |
| FairRF (Zhao et al. (2022)) | $63.26 \pm 0.83\%$ | $25.67 \pm 2.63\%$ | $21.47 \pm 1.76\%$ |
| Chai et al. (2022)(softmax label) | $63.47 \pm 0.44\%$ | $21.32 \pm 1.97\%$ | $19.52 \pm 2.46\%$ |
| Chai et al. (2022)(linear label) | $63.34 \pm 0.46\%$ | $20.31 \pm 2.62\%$ | $20.27 \pm 2.34\%$ |
| Reckoner | $64.00 \pm 0.99\%$ | $17.10 \pm 2.01\%$ | $20.12 \pm 2.20\%$ |
| Reckoner (noise) | $62.68 \pm 1.05\%$ | $16.91 \pm 1.91\%$ | $19.77 \pm 1.86\%$ |
| Reckoner (*pseudo-learning*) | $64.17 \pm 0.94\%$ | $18.38 \pm 1.05\%$ | $21.57 \pm 1.01\%$ |

Table 3: Results on the New Adult dataset

| Metrics(%) Methods | Accuracy | Equalised Odds | Demographic Parity |
|---|---|---|---|
| DRO (Hashimoto et al. (2018)) | $84.51 \pm 0.26\%$ | $11.59 \pm 1.67\%$ | $12.07 \pm 1.50\%$ |
| ARL (Lahoti et al. (2020)) | $85.32 \pm 0.32\%$ | $12.11 \pm 1.30\%$ | $13.41 \pm 1.06\%$ |
| FairRF (Zhao et al. (2022)) | $83.74 \pm 0.86\%$ | $11.23 \pm 1.42\%$ | $11.37 \pm 1.46\%$ |
| Chai et al. (2022)(softmax label) | $84.63 \pm 0.47\%$ | $10.34 \pm 1.22\%$ | $10.63 \pm 1.34\%$ |
| Chai et al. (2022)(linear label) | $84.27 \pm 0.31\%$ | $10.57 \pm 1.64\%$ | $10.21 \pm 1.52\%$ |
| Reckoner | $84.02 \pm 0.06\%$ | $5.33 \pm 1.02\%$ | $8.28 \pm 0.42\%$ |
| Reckoner (noise) | $83.67 \pm 0.54\%$ | $5.41 \pm 2.00\%$ | $7.65 \pm 0.84\%$ |
| Reckoner (*pseudo-learning*) | $83.65 \pm 0.52\%$ | $6.14 \pm 0.63\%$ | $9.18 \pm 0.39\%$ |

best result in Equalised Odds with a relative improvement of about 3.21% over the best baseline Compared to Chai et al. (2022) with optimal Demographic Parity, although our method exhibits a marginal difference of 0.6%, we hold advantages in terms of accuracy and Equalised Odds, with improvements of 0.53% and 4.22%, respectively. In comparison to the highest accuracy achieved by ARL (Lahoti et al. (2020)), we have a significant edge in fairness, with improvements of 3.91% for Equalised Odds and 5.29%. In New Adult dataset, Reckoner exhibits significant improvements in fairness evaluations compared to all the baselines. In comparison to the best-performing baselines in terms of fairness, it achieves a 5.01% improvement in Equalised Odds and a 1.93% improvement in Demographic Parity. While the fairness differentials among other baselines are not particularly obvious, our method demonstrates a more notable enhancement in fairness. Simultaneously, Reckoner achieves the third-highest position in terms of prediction accuracy, with a marginal 1.30% gap compared to the most accurate baseline. This difference may be attributed to the accuracy-fairness tradeoff.

## 5.2 ABLATION STUDY

In our ablation study, we demonstrated the effectiveness of the two components in the proposed framework, Reckoner, and the necessity of combining them. Intuitively, the model that does not incorporate learnable noise but incorporate parameters from *Low-Conf* generators may suffer from an abundance of irrelevant information and decreased predictive performance. On the other hand, the model that does not utilise *pseudo-learning* may lack fairness knowledge from *Low-Conf* generators, potentially leading to unfairness amplification. However, relying solely on *High-Conf* generators for prediction may yield improvements in predictive performance.

**Effect of the learnable noise.** Our model without learnable noise trains both generators using original inputs in the refinement stage. Firstly, we assess the dissimilarity between the original inputs and the reconstructed inputs from both models. Subsequently, we compare the performance of Reckoner with and without learnable noise. As illustrated in Figure 3, We observed that the information generated by the proposed model, in comparison to its variant without learnable noise, is more distinct from the original inputs. In contrast to the variant, the proposed model demonstrates

a substantial increase of 5.77 units in the distance between the reconstructed inputs and the original inputs, indicating a larger divergence between the two. However, as indicated by the accuracy measurements in Table 2, its predictive performance shows a 1.32% improvement over the variant. This suggests that our approach stands out due to the introduction of learnable noise, which changes the distribution of certain features. In a supervised learning context, the model adjusts these feature distributions to better emphasise predictability while fitting the target function. On the other hand, in terms of fairness metrics, the variant outperforms the proposed model. This distinction may be attributed to the learning principles of fair classification reflected in the parameters of the *Low-Conf* generator.

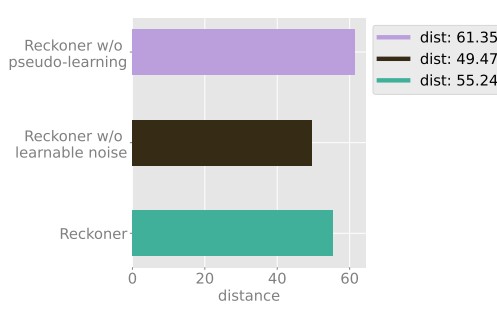

Figure 3: Ablation study on the learnable noise and the *pseudo-learning* using COMPAS dataset.

**Effect of the *pseudo-learning*.** In addition to the learnable noise, we also evaluate the effectiveness of *pseudo-learning*. Our model without *pseudo-learning* is one where the model only utilises the *High-Conf* generator to fit the target function. This variant subsequently trains with new inputs that consist of both learnable noise and original data. From Figure 3, we observe that the proposed model can acquire knowledge about fair classification through *pseudo-learning*. This variant produces the most dissimilar reconstructed inputs compared to the original data but exhibits the best predictive performance (another evidence to the effectiveness of learnable noise). However, its poor fairness performance also underscores the indispensability of *pseudo-learning*.

In contrast to a marginal 0.17% predictive advantage, this variant demonstrates a 1.28% enhancement in unfairness levels compared to our proposed model in Equalised Odds and a 1.45% improvement in Demographic Parity. As hypothesised, the absence of guidance on fairness from the *Low-Conf* generator in this variant's predictions introduces bias errors.

## 6 CONCLUSIONS

In this paper, we present a novel framework for classification tasks that improving fairness without sensitive attributes. Through an analysis of the distribution of non-sensitive attributes across different confidence subsets with respect to different demographic groups, we gain insights into how non-sensitive features are influenced by sensitive attributes and the relationship between fairness and predictability within these subsets. Our proposed framework includes: (1) learnable noise, which is used to force inputs to retain only necessary information for prediction; (2) a dual-model system, employed to enable one model to learn fairness predictions from the other model. Our experimental results show the superiority of the proposed method, which can make accurate and fair predictions, as compared to the state of the art. Our ablation study also confirms the benefits of the two main components in our proposed solution.

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
