# OpenReview forum: "Fairness without Sensitive attributes via Noise and Uncertain Predictions"
_ICLR.cc/2024/Conference — Submitted to ICLR 2024_

### Official Review · Reviewer_zU7R · 2023-10-27

**Soundness:** 2 fair
**Presentation:** 3 good
**Contribution:** 3 good
**Rating:** 5
**Confidence:** 5

**Summary:**

This paper addresses "fairness without demographics" by first splitting the training data into low-confident and high-confident subsets because they observe that low-confident encodes the knowledge of fairness while high-confident is for discriminative information. The refinement is based on dual VAEs in which the high-conf one is to extract most label-related features for classification while simultaneously injecting fairness knowledge from the low-conf one (in an EMA style). Low-conf one is a set unsupervised whose latent representation encouraged to be close to high-conf samples's expectation. They demonstrated the idea on new Adult and COMPAS datasets.

**Strengths:**

1. The overall framework looks good and two stage presentation is clear and logical.
2. The idea of training dual VAEs for two subsets and passing fairness knowledge from low-conf to high-conf is interesting.
3. The improvement on New Adult dataset is significant in terms of EO metric.

**Weaknesses:**

1. The inconsistence between motivation, methodology, and experiments. It will be better if authors can position this paper in a more consistent way.
2. I felt that although authors tried to provide some hints before presenting a component of the method, some of the refinement techniques are still heuristic, which underscores the overall method quality.

I will give the detailed comments in Questions.

**Questions:**

1. In the first paragraph of Introduction, the evaluation to works that maximize the utility of the worst-case group is not precise. The Rawsian fairness (no sensitive attribute) based research leveraged side information (e.g., group ratio) to identify protected group, instead of relying on the correlation with observed features. Following this concern, DRO and ARL used as baselines in this paper are not very suitable, because they both highlighted that accuracy-related utility should be equal across different groups. Note that EO and DP are not criteria designed for these works, although you can conduct so.

2. Given a threshold of 0.6 for splitting the training dataset and the resultant subsets have the distinct performance on EO and DP, which might be questionable. Recalling the definition of EO and DP, we see that they are both computed over predicted y. Since low-conf data tend to appear near to decision boundary, they certainly will yield small EO and DP values. Thus, letting low-conf data represent fairness is not very convincing for me. OOD data is NOT unbiased data.

3. In the refinement stage, notice that only high-conf data will be trained for classification while low-conf examples only contributes some "fair features" in a regularization style. However, from Fig.2, the label in the purple box tried to suggest all data were well mapped to their ground truth labels. So, have you checked if the model has correctly classified low-conf data after training? If yes, why not incorporate a supervised loss therein? If not, how to guarantee a better generalization on test set?

4. As low-conf generator is thought having the desired fairness knowledge, then how about using its mu and sigma as a pseudo supervision (regression) for high-conf data? Any theoretical or experimental evidence of your method? Basically, you are minimizing the distance between any two of N(mu1, sigma1), N(mu2, sigma2), and N(0,I).

5. Generator in the method should refer to the entire VAE. As the final model only takes encoder, can you clarify the role of decoder during training?

6. Two compared methods are from Chai's recent work, leading me to check the connection with this baseline. In Chai's work, they have pointed out the samples near to the decision influence the fairness, while this paper starts from confidence, similar to my insight mentioned above. Authors should clarify their connections.

7. The applied baselines are not very supportive to the targeted challenges. I felt confused why using proxy attributes and fairness-accuracy trade-off works are not included. Also, why only two datasets are used in the paper? Are the proposed method restricted to some specific datasets?

8. Regarding the learnable noise, it is like a patch to this framework, as you have to learn additional model parameters. To make latent features only related to label, one can encourage H(z,y) and reduce H(x,z), from the information theory perspective.

---

> ### Author Response · Authors · 2023-11-23
>
> First of all, we would like to thank you for all these inspiring reviews. All the ideas and hints are very much helpful for improvement of our work in this paper. Especially on the concerns of our motivations and the suggestions on the role of learnable noise. We appreciate the recognition of parts of this work from an expert. Thank you for your contribution to this community. Here are some of our thoughts according to your reviews:
>
> 1. *In the first paragraph of Introduction, the evaluation to works that maximize the utility of the worst-case group is not precise. The Rawsian fairness (no sensitive attribute) based research leveraged side information (e.g., group ratio) to identify protected group, instead of relying on the correlation with observed features. Following this concern, DRO and ARL used as baselines in this paper are not very suitable, because they both highlighted that accuracy-related utility should be equal across different groups. Note that EO and DP are not criteria designed for these works, although you can conduct so.*
> > **1.** Thank you for this correction. Indeed, DRO’s lower bound and upper bound are impacted by group proportion. In the new version, we change to “methods that focus on ensuring that accuracy-related utility is equal across various demographic groups.”
> DRO and ARL used as baselines in this paper are not very suitable
> > **2.** The reason we want to include these two baselines is because they are important methods to deal with fairness without sensitive attributes, though they are in a different branch with us. We self-identify our method as implicit method since we deal with fairness via representation (e.g. mitigating bias from input information). In the new version of paper, we add different marks to better indicate the different types of applied baselines in the table for clearance to our readers.
>
> 2. *Given a threshold of 0.6 for splitting the training dataset and the resultant subsets have the distinct performance on EO and DP, which might be questionable. Recalling the definition of EO and DP, we see that they are both computed over predicted y. Since low-conf data tend to appear near to decision boundary, they certainly will yield small EO and DP values. Thus, letting low-conf data represent fairness is not very convincing for me. OOD data is NOT unbiased data.*
> > **Why showcase the values of EO and DP**？The results presented in the paper regarding EO and DP values aim to provide readers with a more intuitive understanding of how the model tends to be misled by data with high differentiability. Our emphasis is not on highlighting fairness but rather on underscoring the severity of unfairness in the assessments of these data, considered easily differentiable. This becomes evident through a comparison of EO and DP values between these specific data points and the entire dataset. Building on this, our approach develops a framework to correct the model’s biases and alleviate unfairness. In the revised version of the paper, we have modified explanations for certain analytical results.
>  > **Representing fairness with low-confidence data is inaccurate:**  Indeed, the values of EO and DP tend to decrease near the decision boundary. In the new version of paper, we will retain the EO and DP values from high-conf dataset and train_set, and present the number of samples comparison according to different small subsets e.g. # of white vs. # of black with positive labels to better demonstrate our motivations. Since the definition of EO and DP, presenting the number of samples also can help to connect fairness and confidence level.

---

> > ### Author Response · Authors · 2023-11-23
> >
> > 3. *In the refinement stage, notice that only high-conf data will be trained for classification while low-conf examples only contributes some "fair features" in a regularization style. However, from Fig.2, the label in the purple box tried to suggest all data were well mapped to their ground truth labels. So, have you checked if the model has correctly classified low-conf data after training? If yes, why not incorporate a supervised loss therein? If not, how to guarantee a better generalization on test set?*
> > > **1.** Thank you for this review. We check the confusion matrix. Our proposed framework can narrow the gaps of TPR and FPR between Privileged group and Non-privileged group. For low-conf data, we observed that for ours and vanilla ML classifier such as logistic regression, they both tend to predict negative labels when the labels of two classes are balanced (~50% labels of each class). We also observed that even there are more positive class samples belong to high-conf data, our proposed method still tends to predict more negative samples. This may because of the influence from Low-Conf generator.
> > >  **why not supervised loss?** Based on our understanding towards this question, are you asking for an explanation of why not use labels for supervision of Low-Conf generator in the refinement stage? In the new version of the paper, we wil add the results of the experiment using simple two-layer MLP. In this experiment, we used pseudo-labels generated by High-Conf generator to supervise the training of the Low-Conf generator. It is not using ground truth labels but it has supervised loss in this way.
> >
> > 4. *As low-conf generator is thought having the desired fairness knowledge, then how about using its mu and sigma as a pseudo supervision (regression) for high-conf data? Any theoretical or experimental evidence of your method? Basically, you are minimizing the distance between any two of N(mu1, sigma1), N(mu2, sigma2), and N(0,I).*
> > >  Thank you for this suggestion. It is a very interesting idea. We will update the experiment results in the new version of this paper. Also, we consider use Wasserstein distance here instead of Euclidean distance in the future. Thank you very much.
> >
> > 5. *Generator in the method should refer to the entire VAE. As the final model only takes encoder, can you clarify the role of decoder during training?*
> > > We use vae for two main reasons: First reason is its effectiveness presenting through the experiment results and second reason is that we use the decoder’s output, which is the reconstructed input in the ablation study part. Observations through ablation study part help us to understand the reason of using learnable noise and pseudo-learning tricks together.  We admit that there are other model can improve the performance, such as two-layer MLP, which we will also include the experiment results of using MLP in the new version of this paper.
> >
> > 6. *Two compared methods are from Chai's recent work, leading me to check the connection with this baseline. In Chai's work, they have pointed out the samples near to the decision influence the fairness, while this paper starts from confidence, similar to my insight mentioned above. Authors should clarify their connections.*
> > > **1.** Chai’s work is also very interesting. Its motivation/inspiration is from label smoothing. It is different from our motivation, which is from observations from data analysis on feature distribution based on different confidence levels. In the beginning we suspected the biased prediction from overfitting, hence we checked the feature distribution in over-confident samples, but then the observations leaded us to Low-confidence data.
> > >**2.** Chai’s work presents a plot of soft label boundary and hard label boundary, we believe that their work is not very similar to us since we are analysing data from different groups based on confidence levels, not just marginal samples.
> > > **3.** Chai’s work then ensemble the predictions from both teacher and student models (ensembling in their loss function). This way is very natural since it is inspired from label smoothing.  On the other hand, our proposed framework have regularisations on classifier’s weights, aiming to reduce the bias to high-confidence data.
> > > We hope the above can help to address related concerns.
> >
> > 7. *The applied baselines are not very supportive to the targeted challenges. I felt confused why using proxy attributes and fairness-accuracy trade-off works are not included. Also, why only two datasets are used in the paper? Are the proposed method restricted to some specific datasets?*
> > > Thank you for the suggestions. In the future we will include CelebA datasets, an unstructured dataset containing facial images, to better illustrate the generalisation of our framework. As for the baselines, yes we will also include different branches of methods in this paper for better analysis of the performance of this proposed framework.

---

### Official Review · Reviewer_4bz2 · 2023-10-31

**Soundness:** 2 fair
**Presentation:** 1 poor
**Contribution:** 2 fair
**Rating:** 3
**Confidence:** 3

**Summary:**

The paper studies how to mitigate bias when having no access to the full sensitive attributes. The idea is that they find empirically the low-confidence samples (predicted by a classifier) are more biased than the high-confident samples. Therefore, they train a classifier to split the data into low- and high-confidence subsets, and then train VAE on each data together, and the final prediction comes from both VAEs.

**Strengths:**

1. The problem is a practical and important problem given it is harder and harder to access full sensitive attributes

**Weaknesses:**

1. I have trouble understanding the high-level insights of the paper. The authors did not do a good job of presenting their method. e.g. why do they need VAEs? If the idea is to leverage low- and high-confidence data, why not just train two separate classifiers and then use ensemble? What is the purpose of adding learnable noise? I do not understand the author's explanation in 4.3.1. What is $\eta$ in Eq. (1)? What is the corresponding mathematical definition of "Pseudo-distribution" in Figure 1? Is it $\mathcal{L}_L$ in Eq.(2)? In general, I think the technical part is poorly written, and would cause unnecessary confusion to readers.

2. Can authors explain why the design can mitigate bias well when has no access to full sensitive attributes? The sensitive attribute $S$ is rarely mentioned after the problem formulation in Section 3. How does the method connect to missing $S$? I might miss it, but it shows the paper does not highlight how the method works.

3. The experiment on COMPAS does not seem to outperform other methods in an obvious way. The two fairness measures improve but the accuracy also drops. It would be clearer if the results could be presented in an accuracy vs. fairness Pareto frontier style plot.

4. The evaluation is done only on two tabular datasets. This is rare in fairness literature. Can authors justify why it is only tested on two tabular datasets?

**Questions:**

See weakness.

---

> ### Author Response · Authors · 2023-11-23
>
> Thank you for your time.
> 1. *I have trouble understanding the high-level insights of the paper. The authors did not do a good job of presenting their method. e.g. why do they need VAEs? If the idea is to leverage low- and high-confidence data, why not just train two separate classifiers and then use ensemble? What is the purpose of adding learnable noise? I do not understand the author's explanation in 4.3.1. What is 𝛈 in Eq. (1)? What is the corresponding mathematical definition of "Pseudo-distribution" in Figure 1? Is it ℒL in Eq.(2)? In general, I think the technical part is poorly written, and would cause unnecessary confusion to readers.*
> > **Why not ensemble?:** We actually considered this ensemble way in designing the proposed method, but it is not working in the way we want. It makes the prediction more unstable. Preserving the High-Conf’s weights, and adding Low-Conf’s weights, it is like a regulation on High-Conf’s weights, preventing it from getting biased on to those feature distribution patterns of the majority.
> > **Why vae?:** We use vae for two main reasons: First reason is its effectiveness presenting through the experiment results and second reason is that we use the decoder’s output, which is the reconstructed input in the ablation study part. Observations through ablation study part help us to understand the reason of using learnable noise and pseudo-learning tricks together.  We admit that there are other model can improve the performance, such as two-layer MLP, which we will also include the experiment results of using MLP in the new version of this paper. Please, have a look.
>
> 2. *Can authors explain why the design can mitigate bias well when has no access to full sensitive attributes? The sensitive attribute S is rarely mentioned after the problem formulation in Section 3. How does the method connect to missing S ? I might miss it, but it shows the paper does not highlight how the method works.*
> > **1.** To be frank, we are supervised by receiving these requirements of explaining the role of sensitive attributes. We clearly state that sensitive attributes are omitted during the training phase in Section 3 - ‘Problem Definition’, and we also provide motivation for this omission throughout the paper, even the title itself indicates that sensitive attributes are not used in the proposed framework. While it is disappointing to receive such comments, we have added a clear statement at the beginning of the last item in the summarised main points in the Introduction section. We hope that this new statement will help prevent any misunderstanding.
> > **2.** We explain the working flow in the overview part in Part 4 and also in the caption of Figure 2. Please, have a look.
>
> 3. *The experiment on COMPAS does not seem to outperform other methods in an obvious way. The two fairness measures improve but the accuracy also drops. It would be clearer if the results could be presented in an accuracy vs. fairness Pareto frontier style plot.*
> > Thank you for the suggestion. Yes, it is more clear to present an accuracy vs fairness plot in the paper to help our readers, and we will include it in the new version of paper
>
> 4. *The evaluation is done only on two tabular datasets. This is rare in fairness literature. Can authors justify why it is only tested on two tabular datasets?*
> >  We admit more datasets can help readers to understand the permanence of the proposed framework. Hence, we will include new experiments using one new dataset, CelebA for better present our proposed method’s generalisation on different datasets.

---

### Official Review · Reviewer_KsYL · 2023-11-11

**Soundness:** 1 poor
**Presentation:** 3 good
**Contribution:** 2 fair
**Rating:** 5
**Confidence:** 4

**Summary:**

This study tackles the problem of fairness without demographics by initially partitioning the training data into subsets characterized by low and high confidence. The rationale behind this division lies in the observation that low-confidence encodes fairness-related knowledge, while high-confidence pertains to discriminative information. The refinement process employs dual Variational Autoencoders (VAEs), where the high-confidence VAE extracts label-related features for classification. Simultaneously, it injects fairness knowledge from the low-confidence VAE, following an Exponential Moving Average (EMA) style. The low-confidence VAE operates in an unsupervised manner, with its latent representation encouraged to be proximate to the expectation of high-confidence samples. The efficacy of this approach is demonstrated on Adult and COMPAS datasets.

**Strengths:**

1) The problem of fairness without sensitive attributes is important and practical.

2) The writing of this article is well-organized.

**Weaknesses:**

1) (main concern) The proposed method is based on the observation that “when data is close to the decision boundary, non-sensitive information associated with those data tends to be similarly distributed across demographic groups, leading to lower accuracy but increased fairness”. However, it is unclear why and when it happens. For example, does this just happen to be due to the data distribution nature of the COMPAS dataset? The author should provide more explanations and discussions about this, theoretically or empirically, since this key property affects the scope of application of the proposed method.

2) It is also unclear that why the learnable noises need to be added in this paper. Is it to block information of sensitive information? In addition to the final classification results, I suggest that the author give some relevant analytical experiments to prove the validity of the learned noises. For example, if we train a classifier for sensitive attributes on the data with learned noises, is it difficult to predict sensitive attributes accurately?

3) (main concern) It seems that the compared baselines are not strong enough. Please note that some works have gone beyond the baselines used in this paper such as CvaR DRO, LfF, JTT. Moreover, only two tabular datasets are used. I encourage the author to perform experiments on more datasets to illustrate the effectiveness of the proposed method.

**Questions:**

Please refer to Weakness.

---

> ### Author Response · Authors · 2023-11-23
>
> Thank you for the reviews helping to improve our work. We appreciate your suggestions on better data analysis for the paper’s motivation and the interesting experiment on learnable noise for better illustration of its effectiveness.
>
> 1. *(main concern) The proposed method is based on the observation that “when data is close to the decision boundary, non-sensitive information associated with those data tends to be similarly distributed across demographic groups, leading to lower accuracy but increased fairness”. However, it is unclear why and when it happens. For example, does this just happen to be due to the data distribution nature of the COMPAS dataset? The author should provide more explanations and discussions about this, theoretically or empirically, since this key property affects the scope of application of the proposed method.*
> > Thank you for this suggestion. Yes the data analysis is conducted on one dataset, COMPAS. If we have a look on experiment results, the most improvement is in another dataset, New Adult. We argue that this is evidence of good generalisation of our proposed method. However, we appreciate this suggestion, hence we will add data analysis on New Adult in the new version of the paper.
>
> 2. *It is also unclear that why the learnable noises need to be added in this paper. Is it to block information of sensitive information? In addition to the final classification results, I suggest that the author give some relevant analytical experiments to prove the validity of the learned noises. For example, if we train a classifier for sensitive attributes on the data with learned noises, is it difficult to predict sensitive attributes accurately?*
> > Thank you for the suggestion. It is a very interesting idea of training a classifier for sensitive attributes. May you help us to interpret this experiment better? We will be appreciated if we know why this can indicate learnable noises. In our assumptions, we argue the learnable noise is a sort of data augmentation technique, really focusing on how to retain the prediction-only information. Hence, it may also help to predict sensitive attributes and have good prediction results compared to vanilla ML models such as logistic regression.
> In the beginning, we wanted to use learnable noise because of one simple idea. If we have multiple sensitive proxies in the dataset and want to find them all out, one naive way is mask each attribute and train a classifier to see the performance and the fairness evaluation. Currently we have an ablation study result that helps us to understand the effectiveness of learnable noise. We ask the decoder to generate reconstructed input and compare it with the original input. We find if we use learnable noise, although it is very different from the original input, it has the most accurate predictions in all the experiments. Hence we argue the learnable noise is necessary for this proposed framework for accurate predictions.
>
> 3. *(main concern) It seems that the compared baselines are not strong enough. Please note that some works have gone beyond the baselines used in this paper such as CvaR DRO, LfF, JTT. Moreover, only two tabular datasets are used. I encourage the author to perform experiments on more datasets to illustrate the effectiveness of the proposed method.*
> > Thank you for the suggestion. We will include new experiments using one new dataset, CelebA for better present our proposed method’s generalisation on different datasets. For the suggested baselines, we intend to just use DRO and ARL in the comparisons with our work since these two focusing on worst-group performance, ours is not. We include DRO and ARL because we don’t want to miss the two mile stones in this area. Other baselines are chosen from recent research about fairness without missing sensitive attributes/demographic which are representing each different branch of methods. In the new version of paper, we will include proxy method and fairness-accuracy trade-off works.

---

### Meta-Review · Area_Chair_9gEE · 2023-12-12

**Metareview:**

This paper looks at the problem of enforcing/measuring fairness constraints/metrics without access to sensitive group membership for individual inputs.  This is a well-motivated and increasingly well-studied, but largely unsolved, problem.  Most works in this space assume some underlying structure to the relationship between non-sensitive and sensitive attributes, and then work with those models to provide approximate/noisy guarantees around fairness concerns.  This paper does the same, using a novel (to this AC's eyes) approach leaning on VAEs.  Reviewers all appreciated the realistic, timely, and well-motivated application area, and the approach.  Yet, unilaterally there were concerns of presentation, comprehension, reproducibility, and -- to a lesser extent -- comparison against baselines.  Unfortunately, in this AC's eyes, while the work is clearly the foundation of a stronger piece, it can't be adjusted for publication within the timeline for ICLR.

**Justification For Why Not Higher Score:**

Reviewers agreed, even after the rebuttal, that the paper required further work before appearing.

**Justification For Why Not Lower Score:**

N/A

---

### Decision · Program_Chairs · 2024-01-16

Reject